# Acute Hepatopancreatic Necrosis Disease (AHPND): Virulence, Pathogenesis and Mitigation Strategies in Shrimp Aquaculture

**DOI:** 10.3390/toxins13080524

**Published:** 2021-07-27

**Authors:** Vikash Kumar, Suvra Roy, Bijay Kumar Behera, Peter Bossier, Basanta Kumar Das

**Affiliations:** 1Aquatic Environmental Biotechnology and Nanotechnology (AEBN) Division, ICAR-Central Inland Fisheries Research Institute (CIFRI), Barrackpore 700120, India; suvra.roy@ugent.be (S.R.); beherabk18@yahoo.co.in (B.K.B.); basantakumard@gmail.com (B.K.D.); 2Laboratory of Aquaculture & Artemia Reference Center, Department of Animal Sciences and Aquatic Ecology, Faculty of Bioscience Engineering, Ghent University, B-9000 Ghent, Belgium; Peter.bossier@ugent.be

**Keywords:** shrimp aquaculture, AHPND, *V. parahaemolyticus*, virulence mechanism, management strategies

## Abstract

Shrimp, as a high-protein animal food commodity, are one of the fastest growing food producing sectors in the world. It has emerged as a highly traded seafood product, currently exceeding 8 MT of high value. However, disease outbreaks, which are considered as the primary cause of production loss in shrimp farming, have moved to the forefront in recent years and brought socio-economic and environmental unsustainability to the shrimp aquaculture industry. Acute hepatopancreatic necrosis disease (AHPND), caused by *Vibrio* spp., is a relatively new farmed penaeid shrimp bacterial disease. The shrimp production in AHPND affected regions has dropped to ~60%, and the disease has caused a global loss of USD 43 billion to the shrimp farming industry. The conventional approaches, such as antibiotics and disinfectants, often applied for the mitigation or cure of AHPND, have had limited success. Additionally, their usage has been associated with alteration of host gut microbiota and immunity and development of antibiotic resistance in bacterial pathogens. For example, the Mexico AHPND-causing *V. parahaemolyticus* strain (13-306D/4 and 13-511/A1) were reported to carry tetB gene coding for tetracycline resistance gene, and *V. campbellii* from China was found to carry multiple antibiotic resistance genes. As a consequence, there is an urgent need to thoroughly understand the virulence mechanism of AHPND-causing *Vibrio* spp. and develop novel management strategies to control AHPND in shrimp aquaculture, that will be crucially important to ensure food security in the future and offer economic stability to farmers. In this review, the most important findings of AHPND are highlighted, discussed and put in perspective, and some directions for future research are presented.

## 1. Introduction

Crustaceans, usually treated as a subphylum, form a large group of arthropods—mainly aquatic invertebrates—which represent a group of animals important to aquaculture. Crustaceans are considered as economic relevant aquaculture products with high worldwide demand [1,2]. The total crustacean aquaculture production in 2017, from over 30 different species, was 8.4 MT valued at USD 61.06 billion, with an average annual growth rate of 9.92% per year since 2000 [3]. The marine shrimp currently dominate crustacean aquaculture at 5.51 MT or 65.3% of total crustaceans (valued at USD 34.2 billion), followed by freshwater crustacean (2.53 MT or 29.9% total crustacean) and valued at USD 24.3 billion. Although shrimp represents only 6% of the global aquaculture production, they contribute to around 16% of the production value of traded seafood products.

Shrimp production mainly consists of three species, i.e., *Litopenaeus vannamei*, *Penaeus monodon* and *Macrobrachium rosenbergii*. Countries in East and Southeast Asia and Latin America account by far for the major share shrimp production for, but a large proportion of consumption takes place in the developed countries. Among crustaceans, the white leg shrimp (*L. vannamei*) was reported to have the highest unit value at USD 26.7 billion [4]. The giant tiger shrimp *P. monodon* makes up ~15% of total shrimp production, its production reached 0.74 million tonnes, worth USD 5.59 billion [3]. The giant river prawn (*M. rosenbergii*) makes up the rest of the farmed shrimp volumes. *M. rosenbergii* is native to Asia and production reached 0.23 million tons globally, with a value of more than USD 1.90 billion [5,6]. As surveyed in GOAL 2019, the shrimp market is expected to grow further with an annual growth rate of 5.4% between 2017–2021. This will result in a global farmed shrimp harvest of 5.03 million tonnes (approx. 5.4 million tonnes including *M. rosenbergii*) [7,8].

Moreover, as the global human population continues to expand at a high rate and is expected to reach over 9 billion by 2030, shrimp aquaculture can provide global food and nutritional security to people in both developed and developing countries and support the livelihood and jobs of the global population [3,9]. However, due to the global demand increase, the pressure for intensification and expansion of shrimp aquaculture systems has rendered most aquaculture business fragile. In the aquaculture industry, economic losses from disease outbreak have been estimated by the FAO to be over of USD 9 billion per year, which is approximately 15% of the value of world farmed fish and shellfish production. In particular, bacterial diseases have brought socio-economic and environmental unsustainability to the shrimp aquaculture industry during the last decades. Vibriosis, an important bacterial disease, caused by opportunistic *Vibrio* spp. continues as the most serious threat to shrimp farmers in the region [10,11,12,13]. *V. harveyi*, *V. alginolyticus*, *V. anguillarum*, *V. splendidus*, *V. salmonicida*, *V. vulnificus* and *V. parahaemolyticus* strains have been found as main causative organisms of vibriosis [14,15,16]. However, apart from “classical” vibriosis, some *Vibrio* spp. are also responsible for causing acute hepatopancreatic necrosis disease (AHPND), originally known as early mortality syndrome (EMS) [17,18,19]. The AHPND in shrimp aquaculture has escalated since late 2013, when the industry collapsed in South-Asian countries. AHPND, having a devastating impact on the shrimp aquaculture industry, develops quickly, starting approximately 8 days post stocking and severe mortalities (up to 100%) occur within 20–30 days [20,21]. Hence, in this review at first an overview of the current knowledge on acute hepatopancreatic necrosis disease (AHPND) is given, including the disease associated gross signs and histopathology changes. Later, the current status on management/mitigation solutions for acute hepatopancreatic necrosis disease (AHPND) with respect to shrimp aquaculture are summarized.

## 2. Acute Hepatopancreatic Necrosis Disease (AHPND)—An Overview

Acute hepatopancreatic necrosis disease (AHPND), a relatively new farmed penaeid shrimp bacterial disease originally known as early mortality syndrome (EMS) has been causing havoc in the shrimp industry. Since the AHPND outbreak first appeared in China in 2009, it has spread to Vietnam (2010), Malaysia (2011), Thailand (2012), Mexico (2013), Philippines (2015) and South America (2016) (Figure 1) [17,18,19,22]. The shrimp production in AHPND affected regions has dropped temporarily to ~60% and has resulted in collective losses exceeding an estimated USD 43 billion across Asia (China, Malaysia, Thailand, Vietnam) and in Mexico [23,24,25,26,27]. AHPND affects multiple species of shrimp including commercial species, *P. monodon*, *L. vannamei* and *M. rosenbergii* and crustacean model *Artemia franciscana* [24,28,29]. Interestingly, the brine shrimp (*A. franciscana*), an aquatic invertebrate characteristically small, highly osmotolerant and branchiopod crustacean that can be reared under gnotobiotic conditions (allowing full control over the host-associated microbial communities), serve as exceptional model organism to study the host-pathogen interactions in commercially important shrimps and other crustacean species [30,31]. Moreover, the early life stages of shrimp, in general, are more susceptible to AHPND infection. AHPND is characterized by severe atrophy of the shrimp hepatopancreas accompanied by unique histopathological changes at the acute stage of disease [18]. Furthermore, as disease progress massive sloughing of hepatopancreatic or digestive tract epithelial cells in the absence of any accompanying pathogen can be observed within approximately first 30 days of shrimp post-larvae stocking [21,32]. In fact, the AHPND-causing bacteria were reported to mainly target the digestive gland (hepatopancreas) and damage the hepatopancreatic R (resorptive), B (blister), F (fibrillar) and E (embryonic) cells, resulting in dysfunction and massive mortalities of shrimp [19,33,34]. The shrimp affected with AHPND exhibits lethargy, anorexia, slow growth, empty digestive tract and a pale to white hepatopancreas. However, these reported clinical signs for AHPND are common for some other diseases. For instance, the gross signs induced by chemical factor, e.g., nitrite and ammonia or secondary bacterial (traditional vibriosis) and viral (white spot syndrome virus, yellow head virus, etc.) infections could also lead to AHPND pathology [28]. Hence, identification of bacterial virulence factor and AHPND-specific cellular changes coupled with gross clinical signs are considered to be helpful for confirmatory diagnosis of AHPND in shrimp.

### 2.1. Gross Signs and Histopathology of AHPND

The AHPND-affected shrimps are lethargic and display erratic swimming behaviour. The external appearance of shrimps is slightly changed with expanded chromatophores across cuticles. Moreover, based on bacterial density and histological appearance, the natural AHPND-affected shrimp are divided into three phases: (a) initial, (b) acute and (c) terminal phase [35,36].

#### 2.1.1. Initial Phase

The shrimp exhibits signs of damage in the hepatopancreas and in the gut there is partial or total absence of food. Moreover, the changes in the digestive tract and hepatopancreas are visualized better by dissecting and removing epithelial membrane (Figure 2b). The hepatopancreatic tubular epithelial cells are modified and elongated (display drops like appearance) towards the lumen (Figure 3a) causing cellular desquamation. Moreover, as AHPND progresses, the size of hepatopancreatic R and B cells are further reduced (Figure 3b).

#### 2.1.2. Acute Phase

The AHPND-affected shrimp exhibit signs of anorexia and lethargy with empty digestive tract and loss of tissue pigmentation (Figure 2c). The hepatopancreas becomes atrophied and whitish in appearance (Figure 2d). During the first hour of infection, the hepatopancreatic tissue are friable with an aqueous consistency. However, as disease progress the tissue develops a hard consistency, becoming difficult to disintegrate. Microscopically, massive shedding or sloughing of hepatopancreatic tubular epithelial cells are observed. In addition, the tubular epithelial cells are severely necrotized and have massive accumulation of haemocytes and dead cells in the lumen, a pathognomonic lesion reported for AHPND (Figure 3c,d) [17,18,35,37].

Furthermore, at the first hour post-exposure, mitosis is interrupted in hepatopancreatic E cells and the presence of cytoplasmic vacuoles is observed in the B and R cells (Figure 3e). However, as the disease progresses the vacuoles disappear from the hepatopancreatic cells. Interestingly, during the acute phase, no bacterial cells are observed in the AHPND-affected tissue, which suggests that AHPND-causing bacteria secreted binary toxins, i.e., PirA^VP^ and PirB^VP^ might be responsible for mediating AHPND in shrimp at later stage of infection. The PirA^VP^ and PirB^VP^ toxins are found to bind and induce significant damage to the hepatopancreatic tubular epithelial cells, a phenomenon not observed in any other organ or tissue of AHPND-affected shrimp [33,35,38]. At the end of acute phase, the tubular epithelium is severely necrotized, in some instances it is completely absent, and significant high amount of haemocytic infiltration are observed in the intratubular tissues (Figure 3f).

#### 2.1.3. Terminal Phase

Similar to acute phase, the shrimps at terminal phase of AHPND are anorexic, lethargic and have a completely empty stomach. The chromatophores are significantly expanded and hepatopancreas are atrophied and whitish in coloration (Figure 2e). Furthermore, when the hepatopancreas is squashed, it gives a fibrous appearance. Ultrastructure details showed the presence of black streaks, indicating focal melanisation in the hepatopancreatic tubular cells. In addition, the intratubular connective tissue, filled with a large amount of haemocytic cells, are surrounded by haemocytic capsules as a response of bacterial load and necrotic tissue (Figure 3g). During the terminal phase, the damage of tissue is mostly done by PirA^VP^ and PirB^VP^ toxins. However, bacterial proliferation at the site of damage is caused by secondary bacterial infections, possibly by a vibriosis (Figure 3h).

### 2.2. Causative Agent of AHPND

The AHPND is caused by specific strain of bacteria, e.g., *V. parahaemolyticus*, *V. punensis*, *V. harveyi*, *V. owensii*, *V. campbelli* and *Shewanella* sp. that contains pVA1 plasmid (63–70 kb) encoding the binary PirA^VP^ and PirB^VP^ toxins, homologous to the *Photorhabdus luminescens* insect-related (Pir) toxins PirA/PirB (Table 1) [18,39,40,41,42,43]. The PirA^VP^ and PirB^VP^ are the primary virulence factor of AHPND-causing bacteria that mediates AHPND aetiology and mortality in shrimp [20,39,44]

The Photorhabdus insect-related (Pir) toxins are first identified in *Photorhabdus luminescens*, a bacterium that maintains a symbiotic relationship with entomopathogenic nematodes of the family Heterorhabditidae [53,54,55]. In moths and mosquitoes, the binary PirAB toxins, encoded by PirA and PirB genes, are necessary for oral toxicity [56,57]. In fact, during infection, the pathology of oral toxicity can be visualized in the midgut epithelium of moth *Plutella xylostella* larvae, resulting in swelling and shedding of the (apical) epithelial cells [56]. In shrimp aquaculture, the virulent AHPND-causing bacteria containing pVA1 plasmid that encodes PirAB^VP^ toxin genes, homologous to the insecticidal PirA/PirB toxins genes, are absent in all non-AHPND bacterial species [33,58].

Among the PirAB^VP^ binary toxin, PirA^VP^ binds with specific ligands on the cell membrane and receptors (e.g., monosaccharides like N-acetylgalactosamine (GalNAC) and oligosaccharides) and facilitates target-specific recognition. While PirB^VP^ toxin, containing N-terminal domain (PirBN) and C-terminal domain (PirBC), induces cell death in host via pore formation and is involved in protein–protein and protein–ligand interactions [28,59,60,61]. The PirAB^VP^ toxins have been reported to bind with hepatopancreatic epithelial tissue, possibly via recognition and binding to certain ligands on the cell membrane and receptor that leads to oligomerization and pore formation and subsequent cell death [34,37,59]. Interestingly, Kumar et al. (2019) reported that PirAB^VP^ toxins induced focal to extensive necrosis and damage epithelial enterocytes in the midgut and hindgut regions resulting in nuclear pyknosis, cell vacuolisation, mitochondrial and rough endoplasmic reticulum (RER) damage in different degree in gnotobiotic *A. franciscana*. In fact, as the disease proceeded, the epithelium was severely damaged, and the remaining cellular components were further detached into the lumen and showed signs of degeneration such as pyknotic nuclei and lysed cellular membranes, which leads to the subsequent death of challenged larvae (Figure 4). Furthermore, the study showed that PirAB^VP^ toxin affected the digestive process and *A. franciscana* larvae were unable to digest the supplied food [39].

Interestingly, it is important to mention that in some publications it has been proposed that PirB^VP^ toxin alone can induce cell damage and mortality in the host, e.g., mosquito larvae (*Aedes aegypti* and *Aedes albopictus*) and shrimp larvae (*L. vannamei*) [62,63]. The simultaneous occurrence of sloughing and presence of PirB^VP^ toxin in the hepatopancreas provides evidence that PirB^VP^ toxin is enough to cause AHPND infection in shrimp [62]. However, PirA^VP^ and PirB^VP^ toxin mixture was reported to form complex and through receptor binding, oligomerization and pore formation, exhibits a higher toxic effect on experimental animals [19,28,38,53]. Although, PirA^VP^ and PirB^VP^ toxins are directly responsible for shrimp mortality during AHPND [33,59], several other pathogenic extracellular proteins (ECP) are identified in *V. parahaemolyticus* strains like hemin; enterobactin; vibrioferrin; type I, II and VI secretion system protein; chemotaxis protein (60 kDa); flagellin (40 kDa); metalloproteases (PrtV protein, 62 kDa; VppC protein, 90 kDa and VPM protein, 90 kDa); and serine proteases (VPP1, 43 kDa; VpSP37, 37 kDa and PrtA, 71 kda), which might contribute in toxicity of AHPND-causing bacteria [63,64,65,66,67]. For example, the AHPND pathology induced by 1 µg of crude protein (60% ammonium sulfate) precipitated from AHPND-causing *V. parahaemolyticus* broth culture, is equivalent to AHPND caused by 10 µg each of pure PirA^VP^ and PirB^VP^ toxins [28,37]. This indicates that ~10 times more recombinant PirA^VP^ and PirB^VP^ toxin is required to achieve the same results in shrimp larvae caused by crude total protein obtained from *V. parahaemolyticus* AHPND strain. Hence, the AHPND-causing *V. parahaemolyticus* extracellular proteins apparently contain some other toxins or proteins apart from PirA^VP^ and PirB^VP^, which aggravate the toxic effect of PirAB^VP^ toxins [39].

Since, AHPND-causing PirAB^VP^ toxins are released extracellularly, the presence of toxins in the aquatic environment, apart from mediating AHPND and mortality in shrimp (e.g., 20 µg toxin/g shrimp were reported to induce AHPND) [37], may modulate vibriosis, caused by non-AHPND *Vibrio* species. Vibriosis caused by the opportunistic *Vibrio* sp. has negative impacts on fish, crustaceans and molluscs [10,11,12,13]. *V. harveyi*, *V. alginolyticus*, *V. anguillarum*, *V. splendidus*, *V. salmonicida*, *V. vulnificus* and non-AHPND *V. parahaemolyticus* have been found as main causative organisms of Vibriosis [14]. Interestingly, Tran et al. (2020) demonstrated that the presence of PirAB^VP^ toxins modulates the virulence of non-AHPND *Vibrio* spp. in both in vivo and in vitro conditions. The PirAB^VP^ toxin interacts synergistically with *V. harveyi* and *V. alginolyticus* and aggravating vibriosis in a gnotobiotic *A. franciscana* model. However, supplementation of PirAB^VP^ toxin has significant antagonistic interaction on in vivo virulence of *V. campbellii*, *V. parahaemolyticus*, *V. proteolyticus* and *V. anguillarum* strain in the same model [16]. One of the factors that might interfere with virulence of *Vibrio* spp. is the digestive physiology of the host animal [68,69]. The increased virulence might be a result of digestive tract epithelium damage that possibly gives a suitable site of bacterial attachment and further helps in the entry of the pathogen [70]. Kumar et al. (2019) reported that PirAB^VP^ toxins bind with epithelial cells in the midgut and hindgut regions of *A. franciscana* larvae and induce necrosis and damage the cellular structure [32]. Hence, when the *Artemia* larvae were exposed to PirAB^VP^ toxin and *Vibrio* spp. together, the toxin-induced damage of epithelial cells in the digestive tract of larvae might be giving a portal of entry for pathogens, resulting in a significant synergistic increase in vivo virulence of *Vibrio* species. Moreover, the cellular and humoral components of immune system present in the digestive tract play important role in preventing the potential binding and invasion of intestinal layer by an incoming pathogen [71]. As reported in sea bass larvae (*Dicentrarchus labrax*), the gut epithelial enterocytes containing lysosomes mediate intracellular elimination of pathogenic *V. anguillarum* cells [72,73]. The binding of PirAB^VP^ toxin with epithelial cells of the digestive tract, might have induced immunological response in brine shrimp larvae [34,74], which subsequently prevents the attachment and entry of pathogenic bacteria and decreases the in vivo virulence of *Vibrio* species (and hence the antagonistic effect of PirAB toxin on vibriosis caused by certain *Vibrio* species/strains). Therefore, it appears that PirAB^VP^ toxins will not always aggravate vibriosis. Damage of epithelial cells might lead to synergistic effects while an immunological response might result in antagonism, all strains are dependent.

### 2.3. Vibrio parahaemolyticus as a Causative Agent of AHPND

*V. parahaemolyticus* is the predominant species causing AHPND in shrimp [18,45]. *V. parahaemolyticus* is heterogenous Gram-negative, non-spore forming and comma-shaped bacteria with a polar flagellum or with several flagella. This pathogen is part of the autochthonous microflora of estuarine and coastal environments, as well as fish, bivalves and crustaceans in tropical to temperate zones all over the world [75,76]. Apart from fish and shellfish species (including shrimp and molluscs), this bacterium has been isolated from water, sediment, plankton, and marine mammals [77,78]. Moreover, the level of *V. parahaemolyticus* in the environment and in various fish and shellfish species may vary depending on environmental and geographical factors. *V. parahaemolyticus* can thrive in high sodium chloride concentration, ranging from 0.5 to 10% with optimal levels between 1 to 3%, and can grow in moderate temperature (5 to 37 °C) [79].

In shrimp aquaculture, *V. parahaemolyticus* is an important aquatic pathogen and several strains are capable of causing acute hepatopancreatic necrosis disease (AHPND) and other important disease resulting in significant economic losses [38,44,80]. The *V. parahaemolyticus* strains implicated in AHPND are unique in carrying a pVA1 plasmid (70 kb) harbouring the virulence genes, PirA^VP^ and PirB^VP^ encoding the binary PirA^VP^/PirB^VP^ toxins [17]. The pVA1 plasmid are reported to contain 45 open reading frames (ORF) with known functions. These include, five putative transposases, one putative ORF with homology to toxin-antitoxin gene pndA associated with post-segregational killing (PSK) system, operon that encodes proteins (~30% homology) to PirA^VP^ and PirB^VP^ toxins, a cluster of conjugative transfer genes and two plasmid mobilization genes. The PirAB operon has both upstream and downstream transposases, suggesting that the operon can be acquired by lateral gene transfer. The PSK system ensures that only progeny containing the plasmid survive, since the stable PSK mRNA in a plasmid-negative strain will be translated into bactericidal pndA toxin [17].

Since, AHPND-causing pVA1 plasmid reported to contain two plasmid mobilization genes and a group of transfer genes for conjugation [17], the plasmid has been reported to mobilize to other *Vibrio* strains (*V. punensis*, *V. harveyi*, *V. owensii*, *V. campbelli*) and even non-*Vibrio* spp. (*Shewanella* sp.) [41,43,50,52]. These processes explain the huge possibility of conversion from non-pathogenic to pathogenic AHPND strain that positively enhance the spread of AHPND [42]. In addition, the 70-kb AHPND plasmid present in V. parahaemolyticus strains are not clonal, but genetically diverse, suggesting that the virulent plasmid has been acquired from several genotypes of *V. parahaemolyticus* by lateral gene transfer [81]. Recently, it has been shown that *V. parahaemolyticus* harbouring intact pVA1 plasmid and PirAB^VP^ genes (tested positive by PCR), did not produce AHPND-causing PirAB^VP^ toxins. In addition, the AHPND positive strains failed to exhibit characteristic AHPND histopathological lesions and mortality in shrimps [82]. Hence, the virulence of AHPND-causing *V. parahaemolyticus* is reported to depend on the production of secreted proteins, PirAB^VP^ toxins, and not on the copy number of PirA^VP^/PirB^VP^ gene [83].

It is also noteworthy to mention that all human pathogenic *V. parahaemolyticus* strains produce thermostable direct hemolysin (TDH) and TDH-related hemolysin (TRH), as the main virulence factors [79]. The *V. parahaemolyticus* strains possess two sets of type III secretion systems (TTSS); TTSS1 in all strains whereas TTSS2 (α and β, containing tdh and trh genes) only in human pathogenic strains [84]. Moreover, the AHPND-causing *V. parahaemolyticus* strains studied so far are reported to lack the genes for tdh, trh and TTSS2 [50,63]. For example, Chonsin et al. (2016) investigated the conventional virulence factors of human pathogenic *V. parahaemolyticus* strains and AHPND-causing *V. parahaemolyticus* strains. The results showed that none of the AHPND-causing strains possess tdh, trh or TTSS2-related genes of human pathogenic strains [81]. There are two identified types of *V. parahaemolyticus* AHPND bacteria reported based on geographical variations [85]. The *V. parahaemolyticus* AHPND-causing strains from Mexico and Central USA are reported to contain a 4243 bp Tn3-like transposon insert at ORF4, that is not present in the Asian isolates (isolated from China, Vietnam and Thailand) [33]. Moreover, the transposon-like insert is unrelated with AHPND aetiology and shows no difference in the virulence between two AHPND isolates, even if it is found on the virulent plasmid [18]. Recently, González-Gómez et al. (2020) analysed nine AHPND isolates of *V. parahaemolyticus* from Mexico harbouring pVA1 sequences along with 38 previously reported pVA1- harbouring *V. parahaemolyticus*. The AHPND strain nucleotide sequences were clustered into three phylogenetic clades (Latin American, Malaysian, and Cosmopolitan) through pangenomic and phylogenetic analysis. The results highlight that among Latin American and Asian AHPND strains, the main structural difference is the absence of Tn3 transposon in the Asian strains. In addition, some deletion in the PirAB region were also found in two of the Latin American strains. Interestingly, the study demonstrates that diagnosis of AHPND through PirAB toxin gene detection may be inadequate due to structural variability of these genes, as noticed in different isolates [86].

## 3. Control and Management of Acute Hepatopancreatic Necrosis Disease (AHPND)—Current Status

The prophylaxis measures to control AHPND mainly focus on pond management (aeration, feeding, etc.) and disinfections before shrimp post-larvae stocking [87]. However, these approaches are not capable to stop the epidemiological situation once AHPND has emerged in a pond or in neighbouring ponds and hence more effective therapeutic measures are urgently needed to control AHPND in shrimps. The conventional approach applied so far in the mitigation or cure of *V. parahaemolyticus* AHPND strains, such as interrupting feeding or application of antibiotics and disinfectants has had limited success [28]. In addition, due to development of multiple resistance, their usage in the food producing sector is under severe scientific and public scrutiny [88]. For example, the AHPND-causing *V. parahaemolyticus* strain from Mexico (13-511/A1 and 13-306D/4) were reported to carry tetB gene coding for tetracycline resistance gene [89], and *V. campbellii* from China was found to carry multiple antibiotic resistance genes [19], hence the application of traditional methods like antibiotics may be ineffective to control the AHPND in shrimp farming system, especially in the long term.

Most of the therapeutic and control measures developed mainly targets AHPND-causing *V. parahaemolyticus*. However, the presence of AHPND-causing pVA1 plasmid (63–70 kb) encoding the binary toxins named PirA^VP^ and PirB^VP^ in non-*Vibrio parahaemolyticus* and even on non-*Vibrio* species has generated concerns since the management measures used to control a particular AHPND causing bacterial strain may not be useful and can generate unwanted economic pressure to farmers. Therefore, the management measures adopted, based on presence or absence of PirAB^VP^ toxins in the shrimp and aquaculture system, can be more suitable to control and eradicate AHPND from shrimp culture system.

Shrimp, lacking an adaptive immune system, rely on their cellular and humoral components of innate immune responses to combat the invading pathogens, due to which the development of therapeutic agents that enhances the adaptive immune response, e.g., vaccines against infectious disease in shrimp aquaculture had very limited success [28,85]. Therefore, methods that can boost the host innate immune response and enhance disease resistance against diseases have drawn much interest in recent years [2,90]. The disease caused by bacterial pathogens in shrimp farming systems are generally controlled by using appropriate management strategies, including supplementation of immunostimulants, prebiotics, probiotics or phages, maintaining optimum water quality, stocking density, post-larvae quality, aeration and feed quality and quantity [91,92]. However, since the outbreak of AHPND in China in 2009, the research has mainly focused on epidemiological studies including characterization of AHPND aetiological agents and associated pathological changes from various geographical locations. Hence, there is an urgent need to develop promising new methods that can become a potential tool to protect the shrimp against AHPND-causing *V. parahaemolyticus*. Moreover, some studies have reported management strategies to control the disease and possibly prevent AHPND outbreak in shrimp aquaculture. Details of potential therapeutic or control agents are summarized below.

### 3.1. Probiotics

Probiotics have emerged as promising alternatives for improving disease resistance in farmed shrimp against AHPND. The probiotics microbes potentially secrete a wide range of extracellular substances and antimicrobial peptides, which improve feed digestion and absorption, boost shrimp health and immunity, promote shrimp growth and reproduction, and enhance survival upon exposure to pathogenic microorganism (Figure 5) [93]. Moreover, the beneficial effect of probiotic microorganism is generally influenced by several factors related to rearing conditions under larger scale, survival ability until reaching the gastrointestinal tract of the host, method of administration, dosage, probiotic strain and shrimp species [2]. Therefore, before application, attention must be paid in course of selecting an appropriate probiotic strain, since unsuitable strain can negatively impact the colonization, nutrient metabolism and assimilation, growth response, immunomodulation and resistance against pathogenic microorganisms.

Maintaining a biological balance among bacteria and algae in aquaculture ponds and gastrointestinal tract of shrimp is one of the ways to reduce the effect of AHPND in shrimp [94]. Probiotics can participate in establishing a balance of gastrointestinal microbial flora, improving the digestive functions and immune system and increase the survival of *L. vannamei* against the pathogenic *V. parahaemolyticus* AHPND strain [95,96]. Since the AHPND-causing bacteria were reported to infect and damage hepatopancreas, subsequent studies to investigate the effect of probiotics mediated balanced gastrointestinal microbial flora on AHPND bacteria and hepatopancreas morphology add further understanding on probiotics mechanism of action. Moreover, Kewcharoen and Srisapoome (2019) reported that supplementation of *Bacillus subtilis* AQAHBS001 strain through the feed resulted in proliferation and colonization of this strain in gastrointestinal tract of shrimp. Additionally, the shrimp postlarvae exhibited enhanced growth performance and immune gene expression and increased disease resistance against *V. parahaemolyticus* AHPND strain [97]. In another study, Chomwong et al. (2018) found that two lactic acid bacteria (LAB), *Lactobacillus plantarum* SGLAB01 and Lactococcus lactis SGLAB02 strain activate the proPO system, by significantly increasing haemolymph phenoloxidase (PO) activity, improving the survival of *L. vannamei* upon challenge with AHPND-causing *V. parahaemolyticus* [98].

Moreover, several probiotic strains are reported to possess antimicrobial abilities against *Vibrio* species, especially *V. parahaemolyticus*, *V. harveyi* and *V. alginolyticus* [99]. The probiotic bacteria were reported to produce a wide range of extracellular substances such as trypsin, lipase, amylase and antimicrobial substances (e.g., bacteriocins and hydrogen peroxide), against a variety of bacterial pathogenic factors [100,101]. For instance, *Bacillus*, *Lactobacillus*, *Rhodopseudomonas* and *Pseudoalteromonas* probiotic strains are reported to inhibit the activity of pathogenic AHPND-causing bacteria by producing inhibitory compounds, one of the mechanisms of action of probiotics [95,96,99]. A total of 19 lactic acid bacteria (LAB), isolated from *L. vannamei*, were characterized based on morphological, biochemical, sequencing techniques and analysed for their ability to inhibit the AHPND-causing *V. parahaemolyticus* strain. The results showed that 3 among 19 isolated LAB strains have the highest antagonizing ability against AHPND *V. parahaemolyticus* strain in vitro, generating inhibition zones ranging from 18 to 20 mm in diameter. In addition, the shrimps fed with LAB supplemented diets displayed significantly higher survival (approximately 80%) upon AHPND *V. parahaemolyticus* challenge [102]. Recently, Wang et al. (2020) demonstrated that the natural product amicoumacins A purified from the cell-free supernatant of *Bacillus subtilis* BSXE-1601 strain harboured acillibactin, fengycin, surfactin, bacilysin and subtilosin A, which are responsible for both in vitro and in vivo anti-*Vibrio* activity against AHPND-causing *V. parahaemolyticus* strain in *L. vannamei* [103].

### 3.2. Phage Therapy

Bacteriophages are viruses, discovered for the first time over 100 years ago in bacterial host by Twort et al. (1915), with dsRNA, ssRNA, dsDNA and ssDNA genome, that can infect prokaryotic organism [104,105]. The bacteriophages are abundant in nature and have been found in both terrestrial and aquatic environment (non-polluted waters, 2 × 10^8^ bacteriophage/mL), and in association with plants and animals [106]. Phages have been proposed as potential management strategy to control infectious disease in both human and animals [107]. The life cycle of bacteriophages includes either a lytic stage (bacteriolytic) or a lysogenic stage (Figure 6). Since, the emergence of bacterial antibiotic resistance problem in animals and humans, the use of phages as a therapeutic agent (shows an effective bacteriolytic activity) is advantageous as it is natural and relatively inexpensive, without serious or irreversible side effects reported to date [107,108,109]. In shrimp aquaculture, the use of phage therapy is well-documented; work is still going on to develop a commercial phage product for shrimp aquaculture. Bacteriophages used in shrimp bacterial pathogens may belong to the family Siphoviridae or Myoviridae [110,111]. In general, the family Siphoviridae member bacteriophages are reported to be lytic phages [112]. For instance, Yang et al. (2020) found that lytic bacteriophages, namely vB_VpS_BA3 and vB_VpS_CA8 (belong to the Siphoviridae family), isolated from sewage were capable of killing the multidrug resistant *V. parahaemolyticus* and hence its use was suggested as a potential biological control agent [113].

In a study by Vinod et al. (2006), the bacteriophage treatment was found to improve the survival of giant tiger prawn, Penaeus monodon, larvae and postlarvae against *Vibrio*-induced luminous bacterial disease [114]. In another study, bacteriophages are reported to control the growth of pathogenic *V. harveyi* and improve the survival of *P. monodon*, against luminous bacterial disease [115]. These studies showed that bacteriophages can be promising alternatives strategies for effective shrimp larval health management and disease control.

Until now, there are only few attempts have been made to control AHPND in shrimp using bacteriophages. Jun et al. (2016) studied bacteriolytic activity of phage pVp-1 (family Siphoviridae phage) against AHPND-causing *V. parahaemolyticus* strains, the infectivity was tested against 22 strains from geographically diverse regions (5 Asian types and 17 Mexican types). The results showed that the pVp-1 phage can infect 90.9% (20 strains among 22 strains) of *V. parahaemolyticus* AHPND strains and further demonstrates bacteriolytic activity against three strains, known to be highly pathogenic [116]. In another study, Jun et al. (2018) found that following prophylactic and therapeutic treatment, pVp-1 phage-treated shrimps exhibit significant recovery from AHPND histopathological lesions [108]. These results highlight that phage could be suitable for prophylactic and/or therapeutic use against AHPND-causing *V. parahaemolyticus*.

Overall, these studies suggest that the usage of lytic phages could be a potential approach to combat AHPND-causing *V. parahaemolyticus* strains. However, considering that the host range for selected phages was 65–70% and the possibility that bacterial strains may develop resistance [78], phage therapy with a consortium of phages would ensure the efficacy against a wide range of bacterial species/strains reported to cause AHPND in shrimp.

### 3.3. Plant-Derived and/or Natural Compounds

The use of antimicrobial agents in aquaculture could lead to the emergence of resistance in the microorganism. Hence, alternatives are being sought over the last few years and the plant-based compounds are one of the available options for this purpose. Plants are a rich source of bioactive compounds like alkaloids and glycosides and synthesize aromatic compounds mostly phenols or their oxygen substituted derivates that might serve as potential antimicrobial agent to control pathogenic bacterial infection in shrimp aquaculture. For instance, the plant-based products, e.g., essential oils and phenolic compounds have been tested and used as an efficient and alternative treatment against microbial infection in aquaculture [117,118]. The important function of plant-based compounds as antimicrobial includes binding to substrate or metal ions and making them unavailable for microbial pathogens, microbial cell membrane disruption, binding to bacterial cell adhesins or other proteins and inhibiting the binding of bacteria to cell membranes, inactivating the microbial enzymes, blocking the viral cell fusion or adsorption in host cell, etc. Moreover, the natural or plant-based products are preferred because of their biodegradability in the environment, i.e., the residues from plant derived compound treatment tend to be biodegradable in the water whereas, from antibiotics or other chemical treatment. However, the plant-based products (e.g., essential oils) might also have an effect on non-target organism [119,120].

Few studies have reported that natural/plant-based compounds can minimize the effect of pathogen and improves the immune system and survival of shrimp species against the *V. parahaemolyticus* AHPND strain. The rose myrtle, *Rhodomyrtus tomentosa*, seed extract shown significantly high antimicrobial activity against AHPND bacteria. In addition, the extract was found to improve the survival of *L. vannamei* against AHPND-causing *V. parahaemolyticus* strain [121]. Later, a study was carried out to determine the effect of plant extract, *Phyllanthus amarus*, against AHPND-causing *V. parahaemolyticus* strain in white leg shrimp, *L. vannamei*. The results showed that both dried and fresh extract from *P. amarus*, exhibited in vivo antibacterial activity against *V. parahaemolyticus* AHPND strain [122]. In another study, essential oil mixture prepared from 10 plants, i.e., *Lavandula latifolia*, *Pinus sylvestris*, *Jasminum officinale*, *Citrus limon*, *Prunus avium*, *Viola odorata*, *Gardenia jasminoides*, *Cocos nucifera*, *Rosa damascene* and *Eucalyptus globulus*, were tested for anti-*V. parahaemolyticus* activity. The essential oil mixture was found to exhibit antimicrobial activity and significantly improve the survival of *L. vanaamei* against AHPND-causing *V. parahaemolyticus* strain [121]. Moreover, seaweeds are also reported to display antimicrobial activity against bacterial pathogen and possess several health-benefiting properties. The protein extract used from red seaweed, *Gracilaria fisheri*, was evaluated for its anti-bacterial activity and protective role against AHPND-causing *V. parahaemolyticus* strain in white leg shrimp. The results exhibited that protein extract inhibits the growth of virulent *V. parahaemolyticus* strain. In addition, the *G. fisheri* protein extract supplementation significantly improved the survival rate of *L. vannamei* with normalized histological features of hepatopancreas following *V. parahaemolyticus* AHPND strain infection [123]. Furthermore, it has been demonstrated that microalgal-bacterial consortia containing microalgae *Picochlorum* strain S1b and bacteria *Labrenzia* sp. strain 8, *Muricauda* sp. strain 50, or *Arenibacter* sp. strain 61, can significantly inhibit the growth of AHPND-causing *V. parahaemolyticus* strain and increase the survival of *L. vannamei* [124]. A synthetic herbal-based polyphenol compound, pyrogallol, demonstrated to exert high in vitro bactericidal efficacy including increased killing rate and degenerative effects against AHPND *V. parahaemolyticus* cells. The study suggests that pyrogallol based antimicrobial agent could be a promising method to control the AHPND in shrimp producing sectors [125]. Although, the above-mentioned studies have documented that plant-derived compounds exhibit a broad spectrum of pharmacological and health promoting effect, the mechanism of action of these compounds in mediating these effects remain a topic of debate. Therefore, further study to understand the underlying mode of action of these compounds in generating protective responses will be helpful to develop a holistic strategy to control AHPND in shrimp.

#### Immunostimulatory Properties of Plant-Based Compound

The natural products from medicinal plants and marine seaweeds, are considered as potential alternatives for prevention and treatment of AHPND in shrimp. Apart from antiviral, antibacterial and antiparasitic properties, the plant-based compounds are rich in secondary metabolites and phytochemical compounds that play an important role in feed intake and digestibility and improving growth performance and health of shrimp [126,127]. Plant-derived compounds can be administered as a whole plant or parts (leaf, root or seeds) or extract compound, via water routine or feed additive—either singly or as a combination of extract compounds—or even as a mixture with prebiotics or immunostimulants (Figure 7) [128,129,130].

Enhancing the immune system of shrimp has gained considerable attention as a potential method that can contribute to protective immunity and help to fight against diseases. The immunostimulatory activity of plant-based compound are contributed in part by phenolics, alkaloids, terpenoids, essential oils, lectins, polypeptides and polyacetylenes (Table 2). There are several reports, which suggest that treatment of crustacean species (like brine shrimp, *Macrobrachium* spp.) with polyphenols, significantly enhances the innate immune response and provide protection to abiotic (salinity, heat) and biotic (pathogenic bacterial infection) stressors [131,132,133]. In recent years, plant-based compounds are identified to possess the property of inducing heat shock protein within the animal in a non-invasive manner [130,134,135]. These compounds/molecules are also commonly called as heat shock protein inducers (Hspi) [136]. Functionally, the protective function of Hsp70 is documented to be due to its molecular chaperone activity maintaining protein homeostasis by protecting the nascent polypeptides from misfolding, facilitating co- and post-translational folding, assisting in assembly and disassembly of macromolecular complexes and regulating translocation [134]. Additionally, Hsp70 is also reported to confer thermal resistance, protect against osmotic stress, prevent oxidative toxicity and damage and improve tolerance against microbial infection [137,138,139]. These observations suggest that HSP plays important role in host immunity and health. Hence, natural compounds/molecules can be used to induce Hsp70 production in host and provide protection against biotic and abiotic stress.

Recently, it has been demonstrated that polyphenol plant-based compound (phloroglucinol) is a potent in vivo enhancer of Hsp70, and this effect mediates induction of resistance in brine shrimp and *Macrobrachium* larvae against AHPND-causing *V. parahaemolyticus* M0904 strain. The ability of polyphenol plant-based compounds to induce resistance in the host and prevent microbial infection has been described to be functionally dependent on antioxidant property, pro-oxidant activity and anti-microbial effects [140,141]. Similarly, the phloroglucinol-induced protective effect in brine shrimp larvae against *V. parahaemolyticus* were found to be linked to its pro-oxidant activity (e.g., generation of hydrogen peroxide, H_2_O_2_). The pro-oxidant action was linked with increased Hsp70 protein production, which stimulate the immune response and induce resistance in brine shrimp and *Macrobrachium* larvae against AHPND-causing *V. parahaemolyticus* strain [24,127].

Though, the plant-derived compounds are reported to improve immunity and health of shrimp, some of them are known to carry toxicological properties as well. Few studies indicate that plants used a food source, may have mutagenic or genotoxic potential [142]. The toxicology of plants may originate from chemical compounds originated from either leaf, root or seeds [143]. Hence, before application, the investigation of optimum dose requirements in different species and life stages, mode of application (immersion, feed or injection) and residual effects on non-target species must be carried out in order to achieve a safe treatment with plant products.

### 3.4. Environmental Manipulation

Aquatic bacteria are often subjected to fluid shear and hydrodynamic forces, created by either natural factors or anthropogenic activities such as the use of aerators and pumping devices frequently used to enhance shrimp productivity [144,145]. Moreover, the microorganisms, including both single-celled and multi-cellular, have evolved to survive in variables and at times extreme conditions and by changing the phenotype it senses and mount effective response to environmental heterogeneity [146,147,148]. Interestingly, *V. parahaemolyticus* cells are capable of replicating in less than ten minutes, as compared to other *Vibrio* species takes over one hour [75,149]. Hence, any change in environmental condition might triggers phenotype switching in *V. parahaemolyticus* that could affect the biological features and induce remodelling of transcription and translational networks requiring to adapt and maintain cellular status. The *V. parahaemolyticus* cells incubated at constant agitation of 110 rpm (called M0904/110) were demonstrated to develop cellular aggregates or floccules and exhibited significantly higher EPS and biofilm formation (~4 folds). In addition, at M0904/110, the cells produce levan and develop purple colonies. However, cells grown at 120 rpm (called M0904/120) did not produce floccules, had lower EPS and biofilm formation, and produce orange-red colonies. Hence, a critically low shaking frequency might favour the production of self-aggregating biofilm in M0904 like it was described in other species such as *Pseudomonas* [150]. Furthermore, the study revealed that AHPND-causing *V. parahaemolyticus* strain, under differential flow conditions (low fluid shear stress) switches to biofilm phenotype causing a major shift in the protein secretome, e.g., alkaline phosphatase PhoX is produced instead of PirA^VP^/PirB^VP^ toxins [151]. Since, the virulence of AHPND strains is reported to be mediated by the production of PirAB^VP^ toxins, the decreased production of virulence related genes including Pir toxins at M0904/110 (biofilm phenotype) results in significantly reduced virulence of AHPND *V. parahaemolyticus* strain in the host model species, i.e., brine shrimp (*A. franciscana*) and freshwater prawn (*M. rosenbergii*) [44]. The study highlights that AHPND *V. parahaemolyticus* strain has two phenotypic forms (virulent and non-virulent) and shaking condition determines the existence of phenotypic form. Hence, designing methods that can induce phenotype switching in AHPND-causing *V. parahaemolyticus* in an aquaculture setting will open the possibility for effective management of AHPND in shrimp farming, without necessarily removing the AHPND-causing bacteria from the culture system.

#### 3.4.1. Biofloc Technology

Growing shrimp in a biofloc system can be a promising alternative strategy to improve environmental conditions and health status of cultured animals. The basic principle of the biofloc system is to recycle waste nutrients, in particular, inorganic nitrogen resulting from uneaten feed and faeces into microbial biomass, that can be used in situ by the cultured animals or be harvested and processed into feed ingredients [152,153,154,155]. In fact, the metabolic processes and biochemical transformations take place directly in the water column, which promotes the overall balance of the system and the health of the farmed shrimp [156]. The heterotrophic microbiota is stimulated by steering the C/N ratio of the water through the modification of carbohydrate content in feed or by addition of a carbon source in the water, so that bacteria can assimilate the waste ammonium for new biomass production [155]. Hence, ammonium/ammonia can be maintained at a low and non-toxic concentration so that water replacement is no longer required (Figure 8). For instance, Avnimelech (2007) noted that the use of biofloc in intensive tilapia culture significantly improved the nitrogen recovery from 23% to 43% [153].

The biofloc are rich in free amino acids such as alanine, glutamate, arginine and glycine, which are reported to serve as diet attractants for shrimp [157]. Hence, it is noted that shrimp in biofloc system consume up to 29% flocculating particles of their daily feed intake [158]. Moreover, apart from serving as protein and lipid sources these aggregates flocs can contain microbe-associated molecular pattern (MAMP) and microbially bioactive components such as carotenoids, vitamins, glutathione, antioxidants and minerals, which nutritionally modulate the shrimp health and immune response and result in better growth performance and increased resistance against pathogenic microbial infections (Figure 8) [157,159,160,161]. For instance, in situ utilization of microbial flocs in biofloc system by aquaculture organism as well as the utilization of processed biofloc as a feed ingredient has been reported to improve growth performance and the health of shrimp [161,162,163,164,165,166]. There are few reports that have illustrated the role of biofloc in stimulating the non-specific immunity and resistance of shrimp against microbial pathogens including AHPND *V. parahaemolyticus* [167,168]. Hostin et al. (2019) designed an experiment to investigate the effect of autotrophic (with or without probiotics) and heterotrophic bioflocs (with or without probiotics) on *L. vannamei* against AHPND bacterial strain. The results showed that heterotrophic bioflocs (with and without probiotics) and autotrophic bioflocs (with probiotics) can decrease the impact of AHPND-causing *V. parahaemolyticus* and the highest survival of *L. vannamei* was observed when challenged in the presence of their respective biofloc suspensions (shrimp grown in biofloc environment but challenged in clear water were not protected). So, the protective effect in shrimp was depending on operational parameters of the biofloc system, namely C/N ratio [91]. Recently, Kumar et al. (2020) demonstrated that the biofloc system regulates the expression of bacterial virulence genes resulting in enhanced survival of *L. vannamei* upon AHPND-causing *V. parahaemolyticus* challenge. The study showed that, in the biofloc system, AHPND-causing *V. parahaemolyticus* possibly switch from free-living virulent planktonic phenotype to a non-virulent biofilm phenotype, as demonstrated by a decreased transcription of flagella-related motility genes (flaA, CheR and fliS), Pir toxin (PirB^VP^) and AHPND plasmid genes (ORF14) and increased expression of the phenotype switching marker AlkPhoX gene in both in vitro and in vivo conditions [46]. Taken together, the ability of the biofloc system to boost the water quality, growth performance and resistance of *L. vannamei* against *V. parahaemolyticus* AHPND strain makes it a potent aquaculture technology that will be valuable to prevent microbial infection including AHPND and increase the shrimp production with high-density and minimal or no water exchange culture.

#### 3.4.2. Pond Management

The above-mentioned management practices including probiotics, phage therapy and plant-derived compounds have shown promising results to control the outbreak of AHPND in shrimp. However, most of these studies are based on laboratory trials and further validation of dose, route of delivery and associated risk factors are still needed to establish the effectiveness in shrimp farm conditions. Moreover, recently Putth and Polchana (2016) demonstrated that by adopting a better farm management practice, shrimp farmers can control AHPND and avoid production losses. The study showed that pre-stocking and post-stocking measures, including evaluation and screening of the health status of post-larvae, feed quality assessment and disinfection of input materials (e.g., sea water) is helpful to control AHPND in shrimp farms [169].

Apart from management measures, the polyculture system has been identified as a potential strategy to control AHPND in shrimp farms. Tran et al. (2014) studied the effect of polyculture system, including tilapia and *L. vannamei*, in controlling AHPND infection and mortality. The results showed that tilapia induced beneficial algal and bacterial blooms in water, promote healthy and balanced biota communities that confer positive effects in controlling AHPND in shrimps [170]. In another study, Boonyawiwat et al. (2017) evaluated factors related with farm characteristics, farm management, pond and water preparation, feed management, post-larvae and stock management in occurrence of AHPND in shrimp. The results demonstrated that the presence of predator fish, multiple shrimp species or high stocking density in culture system contribute to increased risk of AHPND infections. However, alternative approaches like polyculture, water ageing (≥ 7 days long) and delay in feeding after stocking were likely to promote protection against AHPND in shrimp [171].

## 4. Conclusions and Future Perspective

Shrimp aquaculture is one of the fastest growing food producing sectors in the world. However, an outbreak of acute hepatopancreatic necrosis disease (AHPND) has caused significant economic losses in the shrimp farming industry since 2009. The disease is caused by a specific virulent strain of bacteria, including *V. parahaemolyticus*, *V. punensis*, *V. harveyi*, *V. owensii*, *V. campbelli* and *Shewanella* sp. that contains pVA1 plasmid (63–70 kb) encoding the binary PirA^VP^ and PirB^VP^ toxins. Interestingly, the AHPND affected shrimp show unique histopathological changes, including massive sloughing of hepatopancreatic epithelial cells without any accompanying signs of a pathogen, which demonstrates the involvement of bacterial secreted binary PirA^VP^ and PirB^VP^ toxins in inducing AHPND. Moreover, recent studies have demonstrated that, apart from PirAB^VP^ toxins, the AHPND associated strains have other specific virulence factors that might be involved in virulence of AHPND-causing bacteria and disease pathology. Hence, by using the host-pathogen model, further molecular, microbiological and histopathological studies are still needed for effective characterization of virulence factors of AHPND-causing bacteria and diagnosis of AHPND in shrimp.

In the past, chemical and antibiotics have been commonly used in the shrimp culture system to control bacterial diseases including AHPND. However, the excessive and indiscriminate use of antibiotic has resulted in the development of antibiotic-resistant microbes, which may have potential risks for consumer health globally. Moreover, the above-mentioned management approach discussed in this review, including, probiotics, phage therapy, use of plant-based compounds and environmental manipulation might be applicable in shrimp culture system to control the AHPND (alone or in combination). However, even with these developments, the industry is continuously confronted with the devastating impacts of AHPND. Hence, the quest for alternative methods to control AHPND-causing *Vibrio* spp. is an important challenge for the sustainable development of shrimp aquaculture.

## Figures and Tables

**Figure 1 toxins-13-00524-f001:**
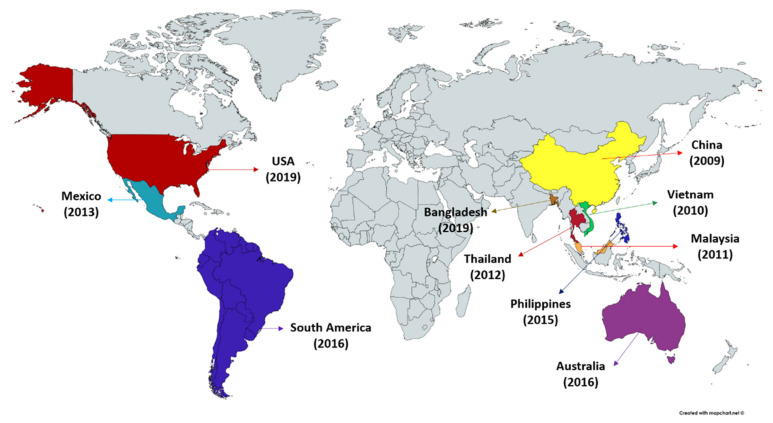
Occurrence of acute hepatopancreatic necrosis disease (AHPND) in shrimp.

**Figure 2 toxins-13-00524-f002:**
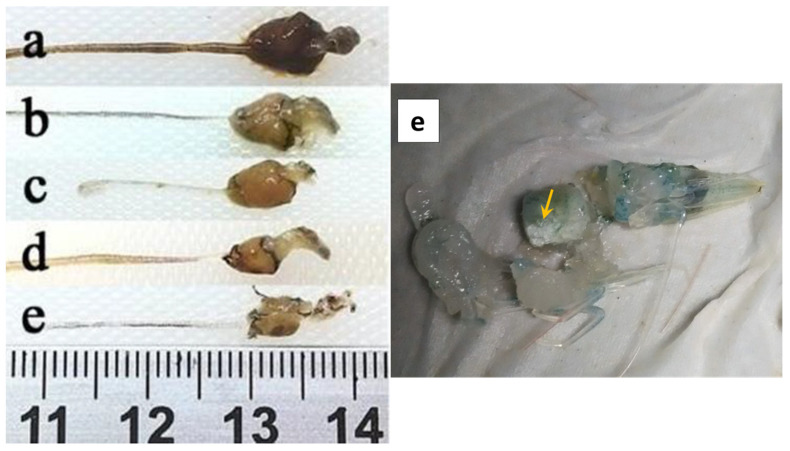
Macroscopic observation of *L. vannamei* digestive tract affected by acute hepatopancreatic necrosis disease (AHPND). (**a**) Healthy shrimp; (**b**) initial phase; (**c**,**d**) acute phase; (**e**) terminal phase. Yellow arrowhead demonstrates completely damaged fibrous appearance hepatopancreas [36].

**Figure 3 toxins-13-00524-f003:**
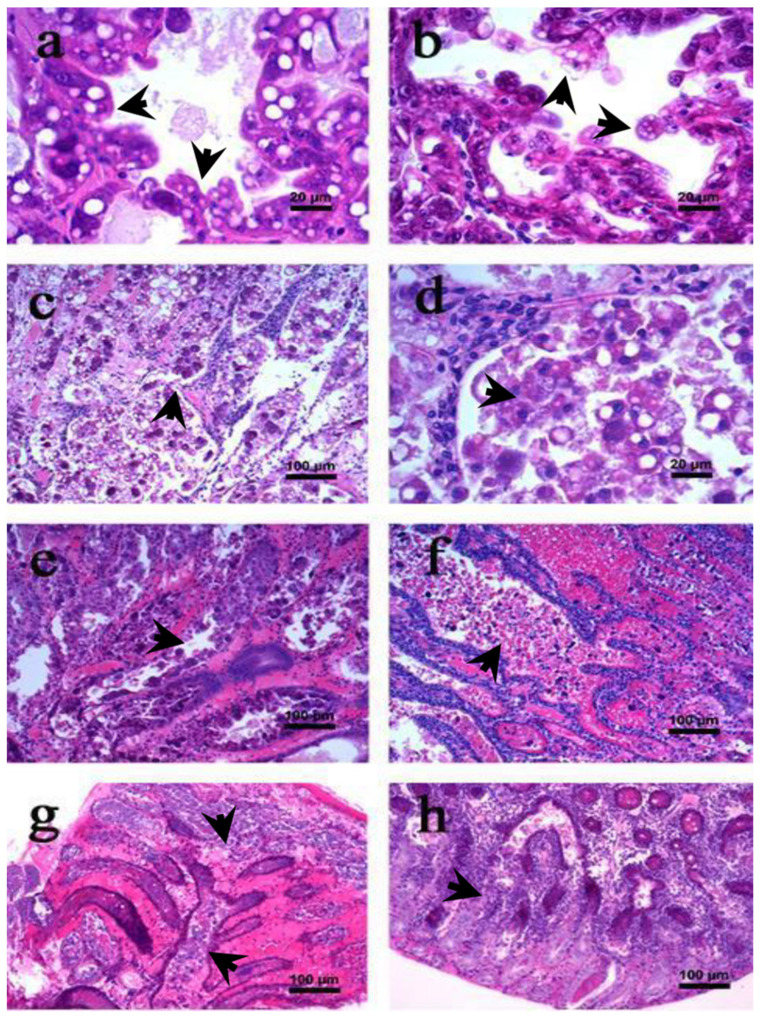
Haematoxylin and eosin (H & E) stained section of hepatopancreas of *L. vannamei* with lesions associated with acute hepatopancreatic necrosis disease (AHPND). (**a**,**b**) Initial phase; (**c**–**f**) acute phase; (**g**,**h**) terminal phase [36]. The arrowhead in figures (**a**–**h**) represent the cellular changes associated different AHPND phases in affected shrimp.

**Figure 4 toxins-13-00524-f004:**
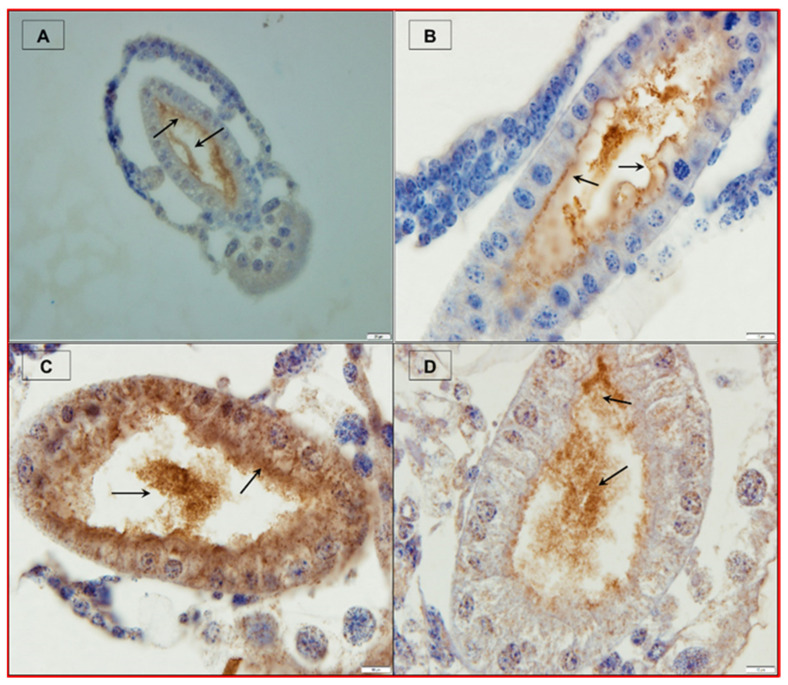
PirAB^VP^ toxin binds the digestive tract and induces sloughing of epithelial cells in brine shrimp larvae. (**A**–**D**) Immunohistochemistry analysis of brine shrimp larvae challenged with PirAB^VP^ toxin. (**A**,**B**): PirAB^VP^ toxin binds to epithelial cells and induces shedding or sloughing of enterocytes in midgut and hindgut digestive tract. (**C**,**D**): Necrosis and damage of epithelial cells and intestinal lumen filled with moderately electron dense cells. (**1**–**4**) Transmission electron microscopy (TEM) analysis of control and treatment group brine shrimp larvae. (**1**,**2**): The digestive tract epithelial enterocytes appeared normal with an intact mitochondrion, nucleus, rough endoplasmic reticulum (RER) and intercellular junctions. (**3**,**4**): PirAB^VP^ toxin challenge produce focal to extensive necrosis and damages epithelial cells in midgut and hindgut region. The arrowhead in figures represent the cellular changes associated with AHPND in affected brine shrimp [39].

**Figure 5 toxins-13-00524-f005:**
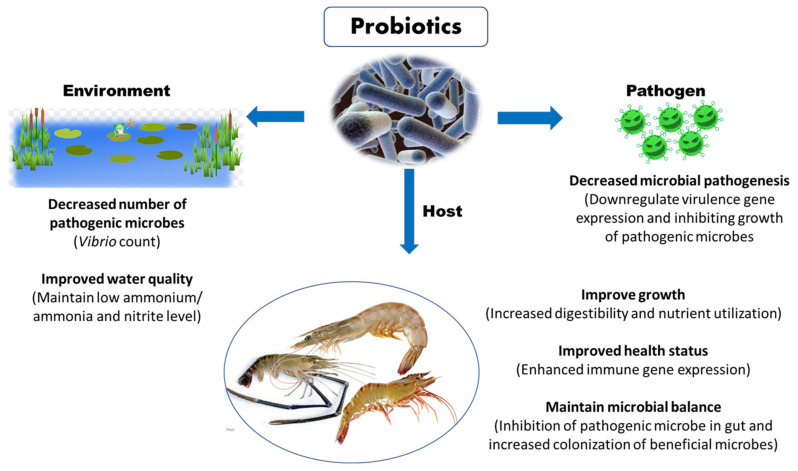
Potential beneficial role of probiotics in shrimp aquaculture.

**Figure 6 toxins-13-00524-f006:**
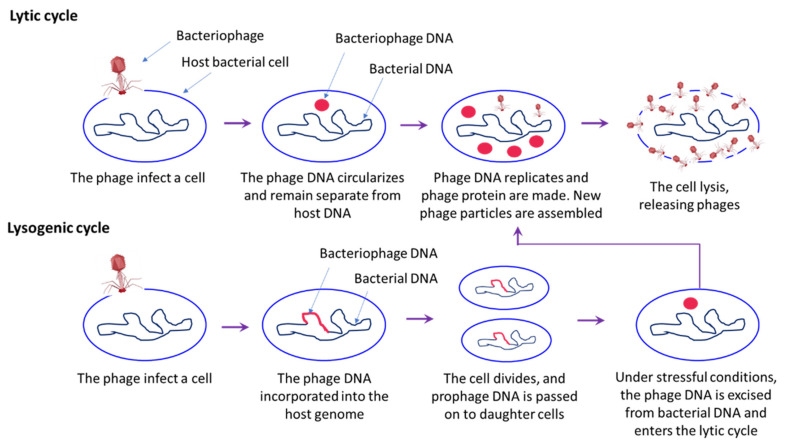
A schematic overview of the bacteriophage life cycle, including lytic and lysogenic cycle. In lytic cycle, bacteriophages infect the host and release of viral genome into bacterial cells. Once a phage infects a bacterium, it shuts down the defence mechanism and takes over its cellular machinery to synthesis new phage particles. The number of phage particles synthesized eventually reaches a point where they rapture the bacterial cells resulting in release of phage particles into the environment that then infects the new host. In lysogenic cycle, phage DNA is incorporated into the bacterial host genome, where it is passed on to the subsequent generations. Environmental stressors such as starvation or exposure to toxic substances may cause the prophage to excise and enter the lytic cycle.

**Figure 7 toxins-13-00524-f007:**
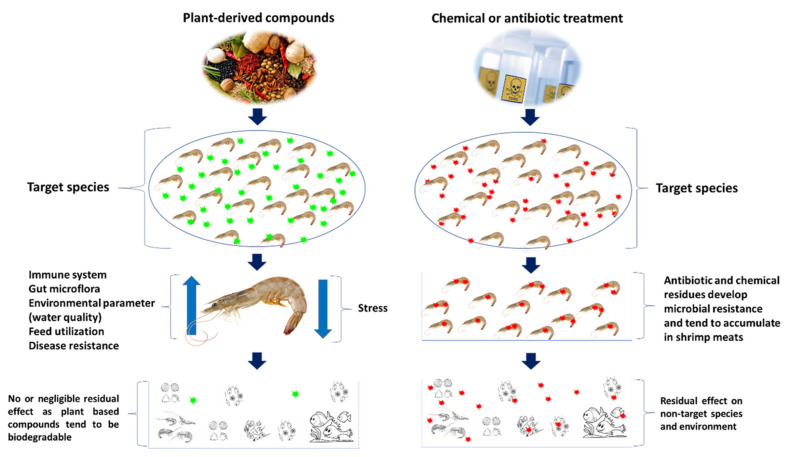
Effect of plant based or natural compounds and conventional compounds in shrimp and environment.

**Figure 8 toxins-13-00524-f008:**
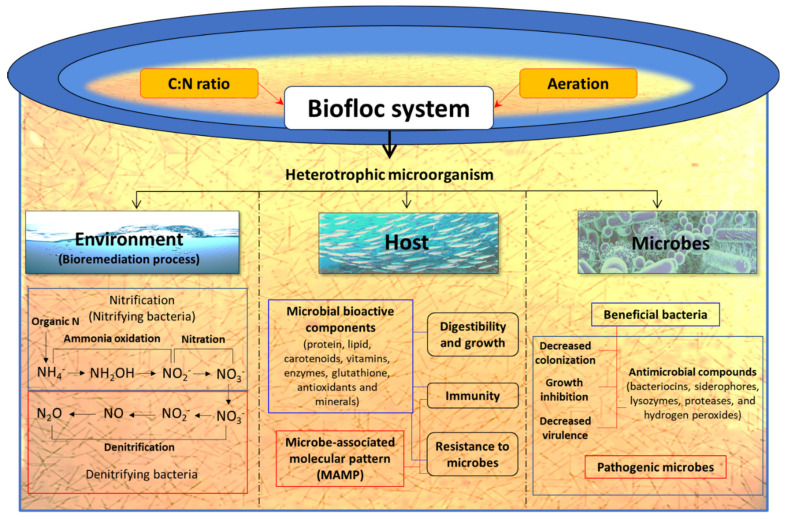
Schematic overview of possible role of biofloc system in host, pathogen and environment in a shrimp aquaculture facility.

**Table 1 toxins-13-00524-t001:** Bacterial species reported to mediate AHPND in crustacean species.

Bacterial Species	Host Range	Geographical Distribution	References
*Vibrio parahaemolyticus*	*P. monodon*, *L. vannamei*	Worldwide	[17,18,33,45,46,47,48,49]
*V. parahaemolyticus*	*Artemia franciscana*	Laboratory condition	[16,24,29,32,39,44]
*V. parahaemolyticus*	*Macrobrachium rosenbergii*	Laboratory condition	[24,44]
*V. punensis*	*L. vannamei*	South America	[42]
*V. harveyi*	*L. vannamei*	China, Malaysia, Vietnam	[42,50,51]
*V. owensii*	*L. vannamei*	China	[41]
*V. campbelli*	*L. vannamei*	China	[40]
*Shewanella* sp.	*L. vannamei*	Thailand	[43,52]

**Table 2 toxins-13-00524-t002:** Role of plant-based compounds in shrimp health.

Class	Chemical Structure	Sub-Class	Example	Role in Aquatic Species
Phenolics	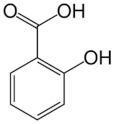	Quinones, lavonoids, flavones, tannins, flavonols	*Allium* spp. (*A. cepa*, *A. sativum*, *A. tuberosum*), *Cynodon dactylon*, *Viscum album*, etc.	Immunostimulant, antioxidant, antimicrobial, growth promotor, anti-helminthic, antiviral
Alkaloids	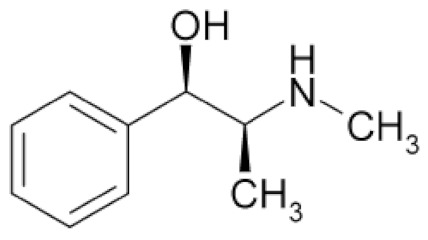		*Camellia sinensis*, *Nicotiana tabacum*, *Aconitum napellus*, *Atropa belladonna*, *Conium maculatum*, etc.	Immunostimulant, antioxidant, antimicrobial, growth promotor, anti-helminthic, antiviral
Terpenoids and essential oils	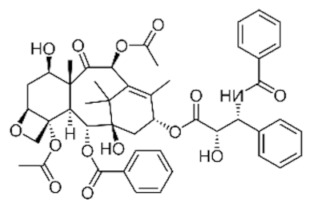		*Pistacia terebinthus*, *Lavandula angustifolia*, *Mentha piperita*, *Melaleuca alternifolia*, etc.	Immunostimulant, antimicrobial, antioxidant, anti-helminthic, growth promotor
Lectins and polypeptides	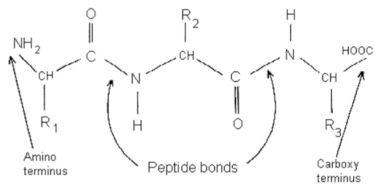		*Glycine max*, *Arachis hypogaea*, *Triticum aestivum*, *Cocos nucifera*, etc.	Antioxidant, antiviral, immunostimulant
Polyacetylenes	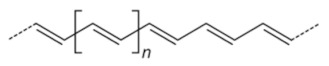 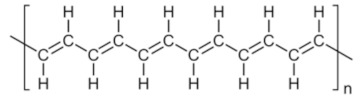		*Anethum graveolens*, *Carum, carvi*, *Daucus carota*, etc.	Immunostimulant, antimicrobial, antioxidant

## Data Availability

Not applicable.

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
