# Peer review of "Acute Hepatopancreatic Necrosis Disease (AHPND): Virulence, Pathogenesis and Mitigation Strategies in Shrimp Aquaculture"

_toxins, 2021, doi:10.3390/toxins13080524_

Round 1
Reviewer 1 Report
The submission is a well organized and thorough review of AHPND and publications reporting treatments and management plans to address the pathology. The review does not get overly involved with the minutia of cited articles and is easily readable with just the right amount of background.
The one criticism is the large number of minor grammar mistakes and a few misspellings. Singular and multiple tenses are often mixed, there are several missing verbs, and subscripts need to be used for chemical nomenclature. Other minor fixes: relay should be changed to rely, the plural of species is spp, not sps, all species names should be in italics. Suggest using grammar check software in Word or other word processor. I do want to mention that the authors do seem to use shrimps (for several species) versus shrimp for singular and plural of the same species properly, which most people get wrong.
Otherwise, the paper does not need any technical revision.
Author Response
Dear reviewer,
We sincerely thank you for your response and comments. We appreciate the thorough review and helpful suggestions, which we believe have contributed to improving the manuscript. We have considered and tried to address all the comments and suggestions given by you.
Furthermore, we have worked on to improve manuscript abstract, methods section, figures and tables. We sincerely hope that the revised version of manuscript will meet the scientific rigor, journal standard and will be considered by Toxins journal.
We have attempted to answer the comments systematically. In the following paragraphs, you will find our response to the comments.
- The one criticism is the large number of minor grammar mistakes and a few misspellings. Singular and multiple tenses are often mixed, there are several missing verbs, and subscripts need to be used for chemical nomenclature. Other minor fixes: relay should be changed to rely, the plural of species is spp, not sps, all species names should be in italics. Suggest using grammar check software in Word or other word processor. I do want to mention that the authors do seem to use shrimps (for several species) versus shrimp for singular and plural of the same species properly, which most people get wrong.
- As suggested by the reviewer, the entire manuscript is thoroughly checked and revised.
Reviewer 2 Report
toxins-1229908
In this review article, the authors cover many important topics related to AHPND, including the symptom details, and the causative agents and the key factors. In addition, the most interesting part for me is that the authors also summarized the potential therapeutic or control agents/methods, such as the use of probiotics, phage therapy, plant-derived and/or natural compounds, and environmental manipulation (e.g. biofloc technology and pond management) to achieve disease control and prevention goals, since antibiotics and disinfectants have limited success in disease control, and are detrimental to the environment and food safety.
Unfortunately however, as presently written, I'm afraid that the language throughout this MS is unacceptably poor, both in terms of coherence and cohesion, with far too many errors to list individually; the entire text is awkward and a struggle to read, and often -- such as eg the very first sentence of the Abstract -- simply ungrammatical.
Other comments:
Production figures in the Abstract (3.5 millions tons [sic]) do not match the numbers given in the Introduction (5.51 MT + 2.53 MT[sic]). The authors also seem to have confused metric tonnes and tons. Lastly, reference [3] is from 2019; it would be better to use the latest figures from the FAO's 2020 report.
Line 87 -- inconsistent reference style.
Fig 1: in what way does this figure illustrate the ‘epidemiology of AHPND’?
Fig 2 has two images labeled ‘e’. What is the meaning of the yellow arrow in the second Fig2 e?
Italics (as for V. parahaemolyticus) and superscripts (as for PirABVP toxin) are not appearing correctly in this pdf.
lines 553-555: figure 7 is referenced to support this sentence, but in fact it seems to be about a great many other things, being primarily a comparison of the effects of plant-derived vs chemical/antibiotic agents.
typos: enahnce, vanaamei
Author Response
Dear reviewer,
We sincerely thank you for your response and comments. We appreciate the thorough review and helpful suggestions, which we believe have contributed to improving the manuscript. We have considered and tried to address all the comments and suggestions given by you.
Furthermore, we have worked on to improve manuscript abstract, methods section, figures and tables. We sincerely hope that the revised version of manuscript will meet the scientific rigor, journal standard and will be considered by Toxins journal.
We have attempted to answer the comments systematically. In the following paragraphs, you will find our response to the comments.
- Unfortunately however, as presently written, I'm afraid that the language throughout this MS is unacceptably poor, both in terms of coherence and cohesion, with far too many errors to list individually; the entire text is awkward and a struggle to read, and often -- such as eg the very first sentence of the Abstract -- simply ungrammatical.
- We appreciate and agree to the comment. As suggested by the reviewer, the entire manuscript is thoroughly checked and revised to avoid grammatical errors and typos.
Specific comments
- Production figures in the Abstract (3.5 millions tons [sic]) do not match the numbers given in the Introduction (5.51 MT + 2.53 MT[sic]). The authors also seem to have confused metric tonnes and tons. Lastly, reference [3] is from 2019; it would be better to use the latest figures from the FAO's 2020 report.
- As suggested by the reviewer, the text has been modified. However, we failed to obtain the latest production figure of crustacean production.
- Line 87 -- inconsistent reference style.
- As per the reviewer suggestions, the cited reference have been modified.
- Fig 1: in what way does this figure illustrate the ‘epidemiology of AHPND’?
- As suggested by the reviewer, the text has been modified in the revised manuscript.
- Fig 2 has two images labeled ‘e’. What is the meaning of the yellow arrow in the second Fig2 e?
- As suggested, the text has been modified in the revised manuscript.
- Italics (as for V. parahaemolyticus) and superscripts (as for PirABVP toxin) are not appearing correctly in this pdf.
- As suggested by the reviewer, the text has been modified in the revised manuscript.
- lines 553-555: figure 7 is referenced to support this sentence, but in fact it seems to be about a great many other things, being primarily a comparison of the effects of plant-derived vs chemical/antibiotic agents.
- As per the reviewer suggestions, the figure 7 has been revised in the manuscript.
Reviewer 3 Report
Need for review is not so clear enough. As most of the information is already published in different format by Food and agricultural organization of united nations. Please see the below link.
http://www.asianfisheriessociety.org/publication/downloadfile.php?id=1217&file=Y0dSbUx6QXhOVFF5TWpNd01ERTFORGMzTXpJM05UZ3VjR1J
Author Response
Dear reviewer,
We sincerely thank you for your response and comments. We appreciate the thorough review and helpful suggestions, which we believe have contributed to improving the manuscript. We have considered and tried to address all the comments and suggestions given by you.
Furthermore, we have worked on to improve manuscript abstract, methods section, figures and tables. We sincerely hope that the revised version of manuscript will meet the scientific rigor, journal standard and will be considered by Toxins journal.
We have attempted to answer the comments systematically. In the following paragraphs, you will find our response to the comments.
Need for review is not so clear enough. As most of the information is already published in different format by Food and agricultural organization of united nations. Please see the below link.
- As suggested by the reviewer, the cited references has been modified in the revised manuscript.